# SaliencyMix: A Saliency Guided Data Augmentation Strategy for Better Regularization

**A. F. M. Shahab Uddin** *
uddin@khu.ac.kr

**Mst. Sirazam Monira** *
monira@khu.ac.kr

**Wheemyung Shin** *
wheemi@khu.ac.kr

**TaeChoong Chung** *†
tcchung@khu.ac.kr

**Sung-Ho Bae**\*†
shbae@khu.ac.kr

## Abstract

Advanced data augmentation strategies have widely been studied to improve the generalization ability of deep learning models. Regional dropout is one of the popular solutions that guides the model to focus on less discriminative parts by randomly removing image regions, resulting in improved regularization. However, such information removal is undesirable. On the other hand, recent strategies suggest to randomly cut and mix patches and their labels among training images, to enjoy the advantages of regional dropout without having any pointless pixel in the augmented images. We argue that such random selection strategies of the patches may not necessarily represent sufficient information about the corresponding object and thereby mixing the labels according to that uninformative patch enables the model to learn unexpected feature representation. Therefore, we propose **SaliencyMix** that carefully selects a representative image patch with the help of a saliency map and mixes this indicative patch with the target image, thus leading the model to learn more appropriate feature representation. SaliencyMix achieves the best known top-1 error of 21.26% and 20.09% for ResNet-50 and ResNet-101 architectures on ImageNet classification, respectively, and also improves the model robustness against adversarial perturbations. Furthermore, models that are trained with SaliencyMix help to improve the object detection performance. Source code is available at https://github.com/SaliencyMix/SaliencyMix.

## 1 Introduction

Machine learning has achieved state-of-the-art (SOTA) performance in many fields, especially in computer vision tasks. This success can mainly be attributed to the deep architecture of convolutional neural networks (CNN) that typically have 10 to 100 millions of learnable parameters. Such a huge number of parameters enable the deep CNNs to solve complex problems. However, besides the powerful representation ability, a huge number of parameters increase the probability of overfitting when the number of training examples is insufficient, which results in a poor generalization of the model.

In order to improve the generalization ability of deep learning models, several data augmentation strategies have been studied. Random feature removal is one of the popular techniques that guides the CNNs not to focus on some small regions of input images or on a small set of internal activations, thereby improving the model robustness. Dropout (Nitish et al., 2014; Tompson et al., 2015) and regional dropout (Junsuk & Hyunjung, 2019; Terrance & Graham, 2017; Golnaz et al., 2018; Singh & Lee, 2017; Zhun et al., 2017) are two established training strategies where the former randomly turns off some internal activations and later removes and/or alters random regions of the input images. Both of them force a model to learn the entire object region rather than focusing on the most

---

*Department of Computer Science & Engineering, Kyung Hee University, South Korea.
†Corresponding author.

| Target Image | Source Image | Augmented Image | |
|---|---|---|---|
|  |  |  |  |
| Mixed label for randomly mixed images | | Dog - 80% & Cat 20% ? | Dog - 80% & Cat 20% ? |

Figure 1: Problem of randomly selecting image patch and mixing labels according to it. When the selected source patch does not represent the source object, the interpolated label misleads the model to learn unexpected feature representation.

important features and thereby improving the generalization of the model. Although dropout and regional dropout improve the classification performance, this kind of feature removal is undesired since they discard a notable portion of informative pixels from the training images.

Recently, Yun et al. (2019) proposed CutMix, that randomly replaces an image region with a patch from another training image and mixes their labels according to the ratio of mixed pixels. Unlike Cutout (Devries & Taylor, 2017), this method can enjoy the properties of regional dropout without having any blank image region. However, we argue that the random selection process may have some possibility to select a patch from the background region that is irrelevant to the target objects of the source image, by which an augmented image may not contain any information about the corresponding object as shown in Figure 1. The selected source patch (background) is highlighted with a black rectangle on the source image. Two possible augmented images are shown wherein both of the cases, there is no information about the source object (cat) in the augmented images despite their mixing location on the target image. However, their interpolated labels encourage the model to learn both objects' features (dog and cat) from that training image. But we recognize that it is undesirable and misleads the CNN to learn unexpected feature representation. Because, CNNs are highly sensitive to textures (Geirhos et al., 2019) and since the interpolated label indicates the selected background patch as the source object, it may encourage the classifier to learn the background as the representative feature for the source object class.

We address the aforementioned problem by carefully selecting the source image patch with the help of some prior information. Specifically, we first extract a saliency map of the source image that highlights important objects and then select a patch surrounding the peak salient region of the source image to assure that we select from the object part and then mix it with the target image. Now the selected patch contains relevant information about the source object that leads the model to learn more appropriate feature representation. This more effective data augmentation strategy is what we call, "**SaliencyMix**". We present extensive experiments on various standard CNN architectures, benchmark datasets, and multiple tasks, to evaluate the proposed method. In summary, SaliencyMix has obtained the new best known top-1 error of 2.76% and 16.56% for WideResNet (Zagoruyko & Komodakis, 2016) on CIFAR-10 and CIFAR-100 (Krizhevsky, 2012), respectively. Also, on ImageNet (Olga et al., 2015) classification problem, SaliencyMix has achieved the best known top-1 and top-5 error of 21.26% and 5.76% for ResNet-50 and 20.09% and 5.15% for ResNet-101 (He et al., 2016). In object detection task, initializing the Faster RCNN (Shaoqing et al., 2015) with SaliencyMix trained model and then fine-tuning the detector has improved the detection performance on Pascal VOC (Everingham et al., 2010) dataset by +1.77 mean average precision (mAP). Moreover, SaliencyMix trained model has proved to be more robust against adversarial attack and improves the top-1 accuracy by 1.96% on adversarially perturbed ImageNet validation set. All of these results clearly indicate the effectiveness of the proposed SaliencyMix data augmentation strategy to enhance the model performance and robustness.

## 2 RELATED WORKS

### 2.1 DATA AUGMENTATION

The success of deep learning models can be accredited to the volume and diversity of data. But collecting labeled data is a cumbersome and time-consuming task. As a result, data augmentation has

been introduced that aims to increase the diversity of existing data by applying various transformations e.g., rotation, flip, etc. Since this simple and inexpensive technique significantly improves the model performance and robustness, data augmentation has widely been used to train deep learning models. Lecun et al. (1998) applied data augmentation to train LeNet for hand written character recognition. They performed several affine transformations such as translation, scaling, shearing, etc. For the same task, Bengio et al. (2011) applied more diverse transformation such as Gaussian noise, salt and pepper noise, Gaussian smoothing, motion blur, local elastic deformation, and various occlusions to the images. Krizhevsky et al. (2012) applied random image patch cropping, horizontal flipping and random color intensity changing based on principal component analysis (PCA). In Deep Image (Wu et al., 2015), color casting, vignetting, and lens distortion are applied besides flipping and cropping to improve the robustness of a very deep network.

Besides these manually designed data augmentations, Lemley et al. (2017) proposed an end-to-end learnable augmentation process, called Smart Augmentation. They used two different networks where one is used to learn the suitable augmentation type and the other one is used to train the actual task. Devries & Taylor (2017) proposed Cutout that randomly removes square regions of the input training images to improve the robustness of the model. Zhang et al. (2017) proposed MixUp that blends two training images to some degree where the labels of the augmented image are assigned by the linear interpolation of those two images. But the augmented images look unnatural and locally ambiguous. Recently, Cubuk et al. (2019) proposed an effective data augmentation method called AutoAugment that defines a search space of various augmentation techniques and selects the best suitable one for each mini-batch. Kim et al. (2020) proposed PuzzleMix that jointly optimize two objectives i.e., selecting an optimal mask and an optimal mixing plan. The mask tries to reveal most salient data of two images and the optimal transport plan aims to maximize the saliency of the revealed portion of the data. Yun et al. (2019) proposed CutMix that randomly cuts and mixes image patches among training samples and mixes their labels proportionally to the size of those patches. However, due to the randomness in the source patch selection process, it may select a region that does not contain any informative pixel about the source object, and the label mixing according to those uninformative patches misleads the classifier to learn unexpected feature representation.

In this work, the careful selection of the source patch always helps to contain some information about the source object and thereby solves the class probability assignment problem and helps to improve the model performance and robustness.

## 2.2 LABEL SMOOTHING

In object classification, the class labels are usually represented by one-hot code i.e., the true labels are expected to have the probability of exactly 1 while the others to have exactly 0. In other words, it suggests the model to be overconfident which causes overfitting to training dataset. As a result, the models have low performance on unseen test dataset. To alleviate this problem, label smoothing allows to relax the model confidence on the true label by setting the class probability to a slightly lower value e.g., lower than 1. As a result, it guides the model to be more adaptive instead of being over-confident and ultimately improves the model robustness and performance (Szegedy et al., 2016). Our method also mixes the class labels and enjoys the benefit of label smoothing.

## 2.3 SALIENCY DETECTION

Saliency detection aims to simulate the natural attention mechanism of human visual system (HVS) and can be classified into two main categories. The first one is a bottom-up approach (Cheng et al., 2014; Zhu et al., 2014; Li et al., 2015; Zhou et al., 2015; Achanta et al., 2009; Li et al., 2013; Hou & Zhang, 2007; Qin et al., 2015; Peng et al., 2016; Lei et al., 2016; Montabone & Soto, 2010) that focuses on exploring low-level vision features. Some visual priors that are inspired by the HVS properties are utilized to describe a salient object. Cheng et al. (2014) utilized a contrast prior and proposed a regional contrast based salient object detection algorithm. Zhu et al. (2014) introduced a robust background measure in an optimization framework to integrate multiple low level cues to obtain clean and uniform saliency maps. Li et al. (2015) optimized the image boundary selection by a boundary removal mechanism and then used random walks ranking to formulate pixel-wised saliency maps. Zhou et al. (2015) proposed a saliency detection model where the saliency information is propagated using a manifold ranking diffusion process on a graph. In addition, some

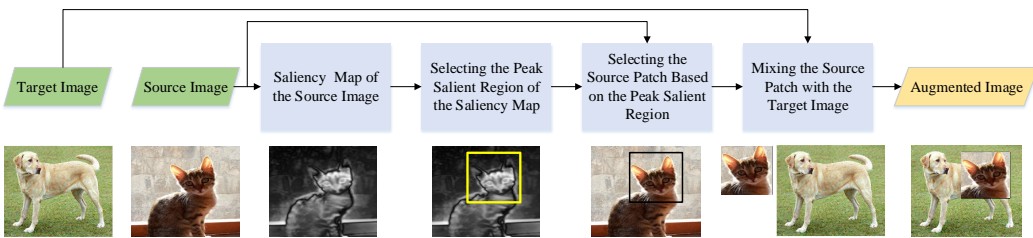

Figure 2: The proposed SaliencyMix data augmentation. We first extract the saliency map of the source image that highlights the regions of interest. Then we select a patch around the peak salient pixel location and mix it with the target image.

traditional techniques are also introduced to achieve image saliency detection, such as frequency domain analysis (Achanta et al., 2009), sparse representation (Li et al., 2013), log-spectrum (Hou & Zhang, 2007) cellular automata (Qin et al., 2015), low-rank recovery (Peng et al., 2016), and Bayesian theory (Lei et al., 2016). Hou & Zhang (2007) proposed a spectral residual method that focuses on the properties of background. Achanta et al. (2009) proposed a frequency tuned approach that preserves the boundary information by retaining sufficient amount of high frequency contents. Montabone & Soto (2010) introduced a method that was originally designed for a fast human detection in a scene by proposing novel features derived from a visual saliency mechanism. Later on this feature extraction mechanism was generalized for other forms of saliency detection.

The second one is a top-down approach which is task-driven and utilizes supervised learning with labels. Several deep learning based methods have been proposed for saliency detection (Deng et al., 2018; Liu et al., 2018; Zhang et al., 2018a;b; Qin et al., 2019). Deng et al. (2018) proposed a recurrent residual refinement network (R3Net) equipped with residual refinement blocks (RRBs) to more accurately detect salient regions. Contexts play an important role in the saliency detection task and based on that Liu et al. (2018) proposed a pixel-wise contextual attention network, called PiCANet, to learn selectively attending to informative context locations for each pixel. Zhang et al. (2018a) introduced multi-scale context-aware feature extraction module and proposed a bi-directional message passing model for salient object detection. Zhang et al. (2018b) focused on powerful feature extraction and proposed an attention guided network which selectively integrates multi-level contextual information in a progressive manner. Recently, Qin et al. (2019) proposed a predict and refine architecture for salient object detection called boundary-aware saliency detection network (BASNet). The author introduced a hybrid loss to train a densely supervised Encoder-Decoder network.

However, despite the high performance of top-down approach, there is a lack of generalization for various applications since they are biased towards the training data and limited to the specific objects. In this study, we require a saliency model to focus on the important object/region in a given scene without knowing their labels. As a result, we rely on bottom-up approach which are unsupervised, scale-invariant and more robust for unseen data. It is worth noting that the training based saliency methods can also be applied where the quality and quantity of data for training the saliency methods may be correlated with the effectiveness of data augmentation. Section 3.3 further explains the effects of different saliency detection algorithm on the proposed data augmentation method.

## 3 PROPOSED METHOD

Similar to Yun et al. (2019), we cut a source patch and mix it to the target image and also mix their labels proportionally to the size of the mixed patches. But in order to prevent the model from learning any irrelevant feature representation, the proposed method enforces to select a source patch in a way so that it must contains information about the source object. It first extracts a saliency map of the source image to highlight the objects of interest and then selects a patch surrounding the peak salient region to mix with the target image. Here we explain the process in detail.

### 3.1 SELECTION OF THE SOURCE PATCH

The goal of saliency detection is to find out the pixels or regions that are attractive to the HVS and to assign them with higher intensity values (Cong et al., 2019). A saliency detection method produces the visual saliency map, a gray-scale image, that highlights the objects of interest and thereby mostly

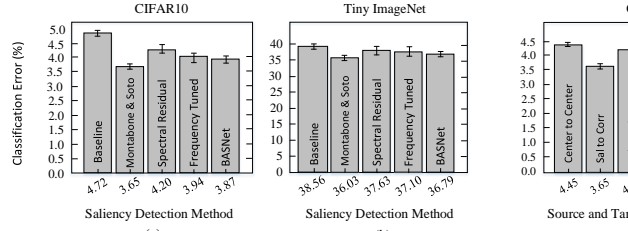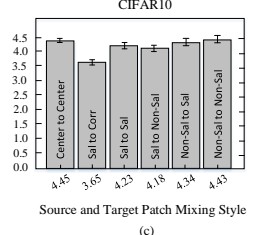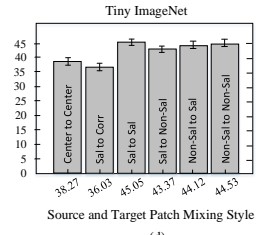

Figure 3: The effect of using different saliency detection methods in the proposed data augmentation on (a) CIFAR10 and (b) Tiny-ImageNet classification tasks. Different ways of selection and mixing of the source patch with the target image and their effects on (c) CIFAR 10 and (d) Tiny-ImageNet classification tasks. Performance is reported from the average of five and three runs for CIFAR10 and Tiny-ImageNet, respectively.

focuses on the foreground. Let $I_s \in \mathbb{R}^{W \times H \times C}$ is a randomly selected training (source) image with label $y_s$ from which a patch will be cut. Then its saliency map detection can be represented as

$$I_{vs} = f(I_s), \tag{1}$$

where $I_{vs} \in \mathbb{R}^{W \times H}$ represents the visual saliency map of the given source image $I_s$ as shown in Figure 2 where the objects of interest have higher intensity values and $f(\cdot)$ represents a saliency detection model. Then we search for a pixel $I_{vs}^{i,j}$ in the saliency map that has the maximum intensity value. The $i, j$ represent the $x$ and $y$ coordinates of the most salient pixel and can be found as

$$i, j = argmax(I_{vs}), \tag{2}$$

Then we select a patch, either by centering on the $I_{vs}^{i,j} - th$ pixel if possible, or keeping the $I_{vs}^{i,j} - th$ pixel on the selected patch. It ensures that the patch is selected from the object region, not from the background. The size of the patch is determined based on a combination ratio $\lambda$ which is sampled from the uniform distribution $(0, 1)$ to decide the percentage of an image to be cropped.

## 3.2 MIXING THE PATCHES AND LABELS

Let $I_t \in \mathbb{R}^{W \times H \times C}$ is another randomly selected training (target) image with label $y_t$, to where the source patch will be mixed. SaliencyMix partially mixes $I_t$ and $I_s$ to produce a new training sample $I_a$, the augmented image, with label $y_a$. The mixing of two images can be defined as

$$I_a = M \odot I_s + M' \odot I_t, \tag{3}$$

where $I_a$ denotes the augmented image, $M \in \{0, 1\}^{W \times H}$ represents a binary mask, $M'$ is the complement of $M$ and $\odot$ represents element-wise multiplication. First, the source patch location is defined by using the peak salient information and the value of $\lambda$ and then the corresponding location of the mask $M$ is set to 1 and others to 0. The element-wise multiplication of $M$ with the source image results with an image that removes everything except the region decided to keep. In contrast, $M'$ performs in an opposite way of $M$ i.e., the element-wise multiplication of $M'$ with the target image keeps all the regions except the selected patch. Finally, the addition of those two creates a new training sample that contains the target image with the selected source patch in it (See Figure 2). Besides mixing the images we also mix their labels based on the size of the mixed patches as

$$y_a = \lambda y_t + (1 - \lambda)y_s, \tag{4}$$

where $y_a$ denotes the label for the augmented sample and $\lambda$ is the combination ratio. Other ways of mixing are investigated in Section 3.4.

## 3.3 IMPACT OF DIFFERENT SALIENCY DETECTION METHODS

Here we investigate the effect of incorporating various saliency detection methods in our Saliency-cyMix data augmentation technique. We use four well-recognized saliency detection algorithms

Table 1: Classification performance (average of five runs) of SOTA data augmentation methods on CIFAR-10 and CIFAR-100 datasets using popular standard architectures. An additional "+" sign after the dataset name indicates that the traditional data augmentation techniques have also been used during training.

| METHOD | TOP-1 ERROR (%) | | | |
| --- | --- | --- | --- | --- |
| | CIFAR-10 | CIFAR-10+ | CIFAR-100 | CIFAR-100+ |
| RESNET-18 (BASELINE) | 10.63± 0.26 | 4.72± 0.21 | 36.68± 0.57 | 22.46± 0.31 |
| RESNET-18 + CUTOUT | 9.31±0.18 | 3.99±0.13 | 34.98±0.29 | 21.96±0.24 |
| RESNET-18 + CUTMIX | 9.44±0.34 | 3.78±0.12 | 34.42±0.27 | 19.42±0.23 |
| RESNET-18 + SALIENCYMIX | **7.59±0.22** | **3.65±0.10** | **28.73±0.13** | **19.29±0.21** |
| RESNET-50 (BASELINE) | 12.14±0.95 | 4.98±0.14 | 36.48±0.50 | 21.58±0.43 |
| RESNET-50 + CUTOUT | 8.84±0.77 | 3.86±0.25 | 32.97±0.74 | 21.38±0.69 |
| RESNET-50 + CUTMIX | 9.16±0.38 | 3.61±0.13 | 31.65±0.61 | 18.72±0.23 |
| RESNET-50 + SALIENCYMIX | **6.81±0.30** | **3.46±0.08** | **24.89±0.39** | **18.57±0.29** |
| WIDERESNET-28-10 (BASELINE) | 6.97±0.22 | 3.87±0.08 | 26.06±0.22 | 18.80±0.08 |
| WIDERESNET-28-10 + CUTOUT | 5.54±0.08 | 3.08±0.16 | 23.94±0.15 | 18.41±0.27 |
| WIDERESNET-28-10 + AUTOAUGMENT | - | **2.60±0.10** | - | 17.10±0.30 |
| WIDERESNET-28-10 + PUZZLEMIX (200 EPOCHS) | - | - | - | **16.23** |
| WIDERESNET-28-10 + CUTMIX | 5.18±0.20 | 2.87±0.16 | 23.21±0.20 | 16.66±0.20 |
| WIDERESNET-28-10 + SALIENCYMIX | **4.04±0.13** | 2.76±0.07 | **19.45±0.32** | 16.56±0.17 |

(Montabone & Soto, 2010; Hou & Zhang, 2007; Achanta et al., 2009; Qin et al., 2019), and perform experiments using ResNet-18 as a baseline model on CIFAR-10 dataset and ResNet-50 as a baseline model on Tiny-ImageNet dataset for 200 epochs and 100 epochs, respectively. Note that the statistical saliency models (Montabone & Soto, 2010; Hou & Zhang, 2007; Achanta et al., 2009) work on any size of images. But the learning based model i.e., Qin et al. (2019) are not scale invariant and it should resizes the input image to $224 \times 224$ and the resulting saliency map is scaled back to the original size. Figure 3(a-b) show that Montabone & Soto (2010) performs better on both the datasets and the effects are identical on CIFA10 and ImageNet datasets. As a result, Montabone & Soto (2010) is used to extract the saliency map in the proposed data augmentation technique.

## 3.4 DIFFERENT WAYS OF SELECTING AND MIXING THE SOURCE PATCH

There are several ways to select the source patch and mix it with the target image. In this section, we explore those possible schemes and examine their effect on the proposed method. We use ResNet-18 and ResNet-50 architectures with SaliencyMix data augmentation and perform experiments on CIFAR-10 and Tiny-ImageNet datasets, respectively. We consider five possible schemes: $(i)$ *Salient to Corresponding*, that selects the source patch from the most salient region and mix it to the corresponding location of the target image; $(ii)$ *Salient to Salient*, that selects the source patch from the most salient region and mix it to the salient region of the target image; $(iii)$ *Salient to Non-Salient*, that selects the source patch from the most salient region but mix it to the non-salient region of the target image; $(iv)$ *Non-Salient to Salient*, that selects the source patch from the non-salient region of the source image but mix it to the salient region of the target image; and $(v)$ *Non-Salient to Non-Salient*, that selects the source patch from the non-salient region of the source image and also mix it to the non-salient region of the target image. To find out the non-salient region, we use the least important pixel of an image.

Figure 3(c-d) show the classification performance of the proposed SaliencyMix data augmentation with the above mentioned selection and mixing schemes. Similar to the Section 3.3, the effects of the proposed method are similar for CIFAR10 and Tiny-ImageNet datasets. Both the *Non-Salient to Salient* and *Non-Salient to Non-Salient* select the source patch from the non-salient region of the source image that doesn't contain any information about the source object and thereby produce large classification error compared to the other three options where the patch is selected from the most salient region of the source image. It justifies our SaliencyMix i.e., the source patch should be selected in such a way so that it must contain information about the source object. On the other hand, *Salient to Salient* covers the most significant part of the target image that restricts the model from learning its most important feature and *Salient to Non-Salient* may not occlude the target object which is necessary to improve the regularization. But *Salient to Corresponding* keeps balance by changeably occluding the most important part and other based on the orientation of the source and target object. Consequently, it produces more variety of augmented data and thereby achieves the lowest classification error. Also, it introduces less computational burden since only the source image saliency detection is required. Therefore, the proposed method uses *Salient to Corresponding* as the default selection and mixing scheme.

Table 2: Performance comparison (the best performance) of SOTA data augmentation strategies on ImageNet classification with standard model architectures.

| Method | Top-1 Error (%) | Top-5 Error (%) |
|---|---|---|
| ResNet-50 (Baseline) | 23.68 | 7.05 |
| ResNet-50 + Cutout | 22.93 | 6.66 |
| ResNet-50 + StochasticDepth | 22.46 | 6.27 |
| ResNet-50 + Mixup | 22.58 | 6.40 |
| ResNet-50 + Manifold Mixup | 22.50 | 6.21 |
| ResNet-50 + AutoAugment | 22.40 | 6.20 |
| ResNet-50 + DropBlock | 21.87 | 5.98 |
| ResNet-50 + CutMix | 21.40 | 5.92 |
| ResNet-50 + PuzzleMix | **21.24** | **5.71** |
| ResNet-50 + SaliencyMix | **21.26** | **5.76** |
| ResNet-101 (Baseline) | 21.87 | 6.29 |
| ResNet-101 + Cutout | 20.72 | 5.51 |
| ResNet-101 + Mixup | 20.52 | 5.28 |
| ResNet-101 + CutMix | 20.17 | 5.24 |
| ResNet-101 + SaliencyMix | **20.09** | **5.15** |

Table 3: Impact of SaliencyMix trained model on transfer learning to object detection task. The results are reported from the average of three runs.

| Backbone Network | ImageNet Cls. Err. Top-1 (%) | Detection (F-RCNN) (mAP) |
|---|---|---|
| ResNet-50 (Baseline) | 23.68 | 76.71 (+0.00) |
| Cutout-trained | 22.93 | 77.17 (+0.46) |
| Mixup-trained | 22.58 | 77.98 (+1.27) |
| CutMix-trained | 21.40 | 78.31 (+1.60) |
| SaliencyMix-trained | **21.26** | **78.48 (+1.77)** |

## 4 EXPERIMENTS

We verify the effectiveness of the proposed SaliencyMix data augmentation strategy on multiple tasks. We evaluate our method on image classification by applying it on several benchmark image recognition datasets using popular SOTA architectures. We also use the SaliencyMix trained model and fine-tune it for object detection task to verify its usefulness in enhancing the detection performance. Furthermore, we validate the robustness of the proposed method against adversarial attacks. All experiments were performed on PyTorch platform with four NVIDIA GeForce RTX 2080 Ti GPUs.

### 4.1 IMAGE CLASSIFICATION

#### 4.1.1 CIFAR-10 AND CIFAR-100

There are $60,000$ color images of size $32 \times 32$ pixels in both the CIFAR-10 and CIFAR-100 datasets (Krizhevsky, 2012) where CIFAR-10 has 10 distinct classes and CIFAR-100 has 100 classes. The number of training and test images in each dataset is $50,000$ and $10,000$, respectively. We apply several standard architectures: a deep residual network (He et al., 2016) with a depth of 18 (ResNet-18) and 50 (ResNet-50), and a wide residual network (Zagoruyko & Komodakis, 2016) with a depth of 28, a widening factor of 10, and dropout with a drop probability of $p = 0.3$ in the convolutional layers (WideResNet-28-10). We train the networks for 200 epochs with a batch size of 256 using stochastic gradient descent (SGD), Nesterov momentum of $0.9$, and weight decay of $5e − 4$. The initial learning rate was $0.1$ and decreased by a factor of $0.2$ after each of the $60, 120,$ and $160$ epochs. The images are normalized using per-channel mean and standard deviation. We perform experiments with and without a traditional data augmentation scheme where the traditional data augmentation includes zero-padding, random cropping, and horizontal flipping.

Table 1 presents the experimental results on CIFAR datasets where the results are reported on five runs average. It can be seen that for each of the architectures, the proposed SaliencyMix data augmentation strategy outperforms all other methods except PuzzleMix (Kim et al., 2020). It is worth noting that PuzzleMix (Kim et al., 2020) and AutoAugment (Cubuk et al., 2019) require additional optimization process to find out the best augmentation criterion, thereby introduces computational burden. On the other hand, rest of the methods do not require such process. The proposed method achieves the best known top-1 error of $2.76\%$ and $16.56\%$ for WideResNet-28-10 on CIFAR-10 and CIFAR-100 datasets, respectively. Moreover, SaliencyMix shows significant performance improvement over CutMix (Yun et al., 2019) when applied without any traditional augmentation technique. It reduces the error rate by $1.85\%, 2.35\%,$ and $1.14\%$ on CIFAR-10 dataset when applied with ResNet-18, ResNet-50 and WideResNet-28-10 architectures, respectively. Using the same architectures, it reduces the error rate by $5.69\%, 6.76\%,$ and $3.76\%$ on CIFAR-100 dataset, respectively.

#### 4.1.2 IMAGENET

ImageNet (Olga et al., 2015) contains 1.2 million training images and $50,000$ validation images of 1000 classes. To perform experiments on ImageNet dataset, we apply the same settings as used in

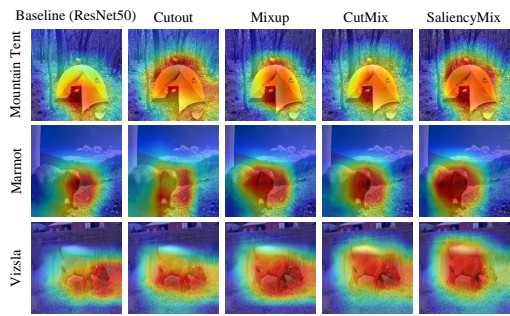

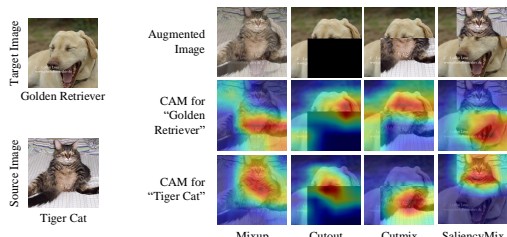

(a) CAM visualizations on un-augmented images. The proposed data augmentation method guides the model to precisely focus on target object.

(b) CAM visualizations on augmented images. Left most column shows the original images, top row shows the input augmented images by different methods, middle and bottom rows show the CAM for 'Golden Retriever', and 'Tiger Cat' class, respectively. The proposed method and Yun et al. (2019) improves the localization ability of the model.

Figure 4: Class activation map (CAM) for models that are trained with various data augmentation techniques. The images are randomly taken from ImageNet (Olga et al., 2015) validation set.

Yun et al. (2019) for a fair comparison. We have trained our SaliencyMix for 300 epochs with an initial learning rate of $0.1$ and decayed by a factor of $0.1$ at epochs $75, 150$, and $225$, with a batch size of 256. Also, the traditional data augmentations such as resizing, cropping, flipping, and jitters have been applied during the training process. Table 2 presents the ImageNet experimental results where the best performance of each method is reported. SaliencyMix outperforms all other methods in comparison and shows competitive results with PuzzleMix (Kim et al., 2020). It drops top-1 error for ResNet-50 by $1.66\%, 1.31\%$, and $0.14\%$ over Cutout, Mixup and CutMix data augmentation, respectively. For ResNet-101 architecture, SaliencyMix achieves the new best result of $20.09\%$ top-1 error and $5.15\%$ top-5 error.

## 4.2 OBJECT DETECTION USING PRE-TRAINED SALIENCYMIX

In this section, we use the SaliencyMix trained model to initialize the Faster RCNN (Shaoqing et al., 2015) that uses ResNet-50 as a backbone network and examine its effect on object detection task. The model is fine-tuned on Pascal VOC 2007 and 2012 datasets and evaluated on VOC 2007 test data using the mAP metric. We follow the fine-tuning strategy of the original method (Shaoqing et al., 2015). The batch size, learning rate, and training iterations are set to $8, 4e-3$, and $41K$, respectively and the learning rate is decayed by a factor of $0.1$ at $33K$ iterations. The results are shown in Table 3. Pre-training with CutMix and SaliencyMix significantly improves the performance of Faster RCNN. This is because in object detection, foreground information (positive data) is much more important than the background (Lin et al., 2017). Since SaliencyMix helps the augmented image to have more foreground or object part than the background, it leads to better detection performance. It can be seen that SaliencyMix trained model outperforms other methods and achieves a performance gain of $+1.77$ mAP.

## 4.3 CLASS ACTIVATION MAP (CAM) ANALYSIS

Class Activation Map (CAM) (Zhou et al., 2016) finds out the regions of input image where the model focuses to recognize an object. To investigate this, we extract CAM of models that are trained with various data augmentation techniques. Here we use a vanilla ResNet-50 model equipped with various data augmentation techniques and trained on ImageNet (Olga et al., 2015). Then we extract CAM for un-augmented images as well as for augmented images. Figure 4 presents the experimental results. Figure 4a shows that the proposed data augmentation technique guides the model to precisely focus on the target object compared to others. Also, Figure 4b shows the similar effect when we search for a specific object in a scene with multiple objects. It can be seen that Mixup (Zhang et al., 2017) has a severe problem of being confused when trying to recognize an object because the pixels are mixed and it is not possible to extract class specific features. Also, Cutout (Devries & Taylor, 2017) suffers disadvantages due to the uninformative image region. On the other hand, both the CutMix (Yun et al., 2019) and SaliencyMix effectively focuses on the corresponding features and precisely localizes the two objects in the scene.

Table 4: Performance comparison on adversarial robustness. Top-1 accuracy (%) of various data augmentation techniques on adversarially perturbed ImageNet validation set.

|  | BASELINE | CUTOUT | MIXUP | CUTMIX | SALIENCYMIX |
|---|---|---|---|---|---|
| ACC. (%) | 8.2 | 11.5 | 24.4 | 31.0 | **32.96** |

Table 5: Training time comparison of various data augmentation techniques using ResNet-18 architecture on CIFAR-10 dataset.

|  | BASELINE | CUTOUT | MIXUP | CUTMIX | SALIENCYMIX |
|---|---|---|---|---|---|
| TIME (HOUR) | **0.83** | 0.84 | 0.87 | 0.89 | 0.91 |

## 4.4 ROBUSTNESS AGAINST ADVERSARIAL ATTACK

Deep learning based models are vulnerable to adversarial examples i.e., they can be fooled by slightly modified examples even when the added perturbations are small and unrecognizable (Szegedy et al., 2014; Goodfellow et al., 2015; Madry et al., 2017). Data augmentation helps to increase the robustness against adversarial perturbations since it introduces many unseen image samples during the training (Madry et al., 2017). Here we verify the adversarial robustness of a model that is trained using various data augmentation techniques and compare their effectiveness. Fast Gradient Sign Method (FGSM) (Madry et al., 2017) is used to generate the adversarial examples and ImageNet pre-trained models of each data augmentation techniques with ResNet-50 architecture is used in this experiment. Table 4 reports top-1 accuracy of various augmentation techniques on adversarially attacked ImageNet validation set. Due to the appropriate feature representation learning and focusing on the overall object rather than a small part, SaliencyMix significantly improves the robustness against the adversarial attack and achieves $1.96\%$ performance improvement over the nearly comparable method CutMix (Yun et al., 2019).

## 4.5 COMPUTATIONAL COMPLEXITY

We investigate the computational complexity of the proposed method and compare it with other data augmentation techniques in terms of training time. All the models are trained on CIFAR-10 dataset using ResNet-18 architecture for 200 epochs. Table 5 presents the training time comparison. It can be seen that SaliencyMix requires a slightly longer training time compared to others, due to saliency map generation. But considering the performance improvement, it can be negligible.

## 5 CONCLUSION

We have introduced an effective data augmentation strategy, called SaliencyMix, that is carefully designed for training CNNs to improve their classification performance and generalization ability. The proposed SaliencyMix guides the models to focus on the overall object regions rather than a small region of input images and also prevents the model from learning in-appropriate feature representation by carefully selecting the representative source patch. It introduces a little computational burden due to saliency detection, while significantly boosts up the model performance and strengthen the model robustness on various computer vision tasks. Applying SaliencyMix with WideResNet achieves the new best known top-1 error of $2.76\%$ and $16.56\%$ on CIFAR-10 and CIFAR-100, respectively. On ImageNet classification, applying SaliencyMix with ResNet-50 and ResNet-101 obtains the new best known top-1 error of $21.26\%$ and $20.09\%$, respectively. On object detection, using the SaliencyMix trained model to initialize the Faster RCNN (ResNet-50 as a backbone network) and fine-tuning leads to a performance improvement by $+1.77$ mAP. Furthermore, SaliencyMix trained model is found to be more robust against adversarial attacks and achieves $1.96\%$ accuracy improvement on adversarially perturbed ImageNet validation set compared to the nearly comparable augmentation method. Considering more detailed and/or high level semantic information for data augmentation will be our future work.

ACKNOWLEDGMENTS

This research was supported by Basic Science Research Program through the National Research Foundation of Korea (NRF) funded by the Ministry of Science, ICT & Future Planning (2018R1C1B3008159) and Basic Science Research Program through the National Research Foundation of Korea (NRF) under Grant [NRF-020R1F1A1050014].

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
