# OpenReview forum: "SaliencyMix: A Saliency Guided Data Augmentation Strategy for Better Regularization"
_ICLR.cc/2021/Conference — ICLR 2021 Poster_

### Official Review · AnonReviewer3 · 2020-10-27
**Interesting trick but not enough for ICLR**

**Rating:** 3
**Confidence:** 3

**Review:**

This paper is based on an interesting observation that previous data augmentation tricks cut&mix may select regions that do not contain useful information. Instead, this paper use saliency models to detect the salient regions first and then cut and mix these salient regions in a source image.

This observation is interesting and worth trying. However, the contribution might be too limited for a ICLR conference paper:
1. as cut&mix cuts bigger blocks in an image, the target object is more likely to be selected. Also for most image classification dataset, the images are quite iconic, so the improvement on classification tasks are limited (as shown in Tab. 1, C10+ and C100+). This might help with detection as it may train models to focus on the most discriminative part of the image, but recent works show that there is no direct correlation between between the performance of the same backbone on detection tasks and classification tasks. In addition, the author didn't provide analysis on what causes the 1.8% improvement on detection tasks (better on smaller objects?) So it's not clear how helpful this trick is.
2. While the method is simple, I expect either some mathematic proof or this method works well on various tasks. The paper didn't have any proof or statistical analysis. This paper didn't either show if the proposed method will work on more tasks (for example segmentation or GAN? the detection provided in this paper is using the backbone initialized from classification, during training faster-rcnn it seems that the trick is not used).

---

> ### Author Response · Authors · 2020-11-19
> **Response to Reviewer#3**
>
> -  We thank the reviewer for the valuable comments.
>
>   In CutMix, the source patch size is defined by a combination ratio lambda which is sampled from a uniform distribution (0, 1) and the cut patch is then mixed to the target image. As a result, it may not guarantee that the selected patch is always bigger enough to contain source object information.
>
>   Regarding the limited performance improvement: From Table 1 of the submitted manuscript it can be seen that when the proposed method was applied without any traditional data augmentation, it significantly improved the classification performance of each model on every dataset. For instance, compared to the nearly competitive method CutMix, the proposed method improves the model performance by 1.85% and 5.69% for ResNet18, 2.35% and 6.76% for ResNet50 and 1.14% and 3.76% for WideResNet on CIFAR10 and CIFAR100 datasets, respectively.
>
>   Also, besides the performance improvement, the proposed method has proven itself to be effective on multiple datasets (CIFAR10, CIFAR100, and ImageNet) and tasks (image classification, object detection, and robustness against adversarially perturbed ImageNet classification).
>
>   Regarding the object detection. We agree that there is no direct relation between the performance of a backbone network and the whole detection network [Ref1]. However, it has been shown that the decent data augmentation method also improves the image detection performance when being used for initializing backbone networks [Ref4]. We believe that this is because good data augmentation methods tend to increase localization ability of neural networks where the localization ability is considered as a key point to solve image detection problems [Ref4]. In order to reflect the reviewer’s comment, we visualize the CAM to qualitatively check the localization ability for models trained with different data augmentation methods. The visualization results show that the proposed SaliencyMix tends to help increasing in localization performance as the CAM maps focus more on target objects (e.g., Mountain Tent, Marmot, and Vizsla etc.). We added the CAM results in Figure 4 of the revised manuscript. In consequence, although we cannot say that CAM results are directly related to the object detection performance, it may provide a clue for linking the relation.
>
> - Regarding the mathematical proof on effective boundary for data augmentation, it has not been carried out in the conventional data augmentation papers [Ref2, Ref3, Ref4]. This is because the data augmentation is a pre-processing technique and there are still unveiled regions for the internal behaviors of deep neural networks. Therefore, the data augmentation methods have advanced based on intuition on data and network properties. Likewise, our work coincides with the conventional ones. However, we performed experiments on object detection and classification on adversarially perturbed ImageNet validation set to verify the generalization ability of the proposed data augmentation method. It is worth noting that we have used pre-trained models (trained on ImageNet classification) to initialize these models. Exploring the applicability of the proposed data augmentation strategy on these experiments will be our future work.
>
>   Also, The reviewer suggests to verify the proposed method on more tasks such as segmentation and GAN. Generally, GANs train themselves to learn the underlying data distribution. The data augmentation, such as Mixup [Ref2], Cutout [Ref3], CutMix [Ref4] and the proposed one, highly alters the data distribution of the training samples. As a result, the model learns to produce unwanted color and geometric distortion (e.g., unnatural color, cutout holes) as introduced by these augmentations and results in a significantly worse performance [Ref5]. On the other hand, for semantic segmentation, we need precise boundary and labels for the objects. Also, the local regions for objects are contextually correlated with neighbor regions. However, this kind of strong augmentation selects some region without being concerned about their boundaries and contexts. As a result, this approach is out of the scope of this study.
>
> References:
>
> [Ref1]. Bochkovskiy, Alexey, Chien-Yao Wang, and Hong-Yuan Mark Liao. "YOLOv4: Optimal Speed and Accuracy of Object Detection." _arXiv preprint arXiv:2004.10934_ (2020).
>
> [Ref2]. DeVries, Terrance, and Graham W. Taylor. "Improved regularization of convolutional neural networks with cutout", _arXiv preprint arXiv:1708.04552 (2017)._
>
> [Ref3]. Zhang, Hongyi, et al. "mixup: Beyond empirical risk minimization", _arXiv preprint arXiv:1710.09412 (2017)_.
>
> [Ref4]. Yun, Sangdoo, et al. "Cutmix: Regularization strategy to train strong classifiers with localizable features", _ICCV,_ 2019.
>
> [Ref5]. Shengyu Zhao, Zhijian Liu, Ji Lin, Jun-Yan Zhu and Song Han, "Differentiable Augmentation for Data-Efficient GAN Training", _NeurIPS,_ 2020.

---

### Official Review · AnonReviewer2 · 2020-10-28
**Simple idea but results are not strong enough**

**Rating:** 3
**Confidence:** 4

**Review:**

This paper proposes a new augmentation method based on CutMix. The authors find out that randomly selecting may mix background textures and this will mislead the model. So, they propose to use saliency maps to control the selection of mixed patches, which is called SaliencyMix. This idea seems easy and reasonable, many experiments are conducted to prove the effectiveness of the proposed method. However, the experiments’ results fail to show the ability of the method, and some explanation is missed.

Positive:
1.     The idea is simple and clear, the paper is well organized and easy to follow.
2.     The experiments are comprehensive, including classification and transfer learning.

Weakness:
1.     The main concern is the effectiveness of the proposed method. According to the authors’ experiments, the improvement over CutMix is very limited on all datasets. Especially on ResNet-101, the promotion over CutMix is only 0.08.
2.     According to the authors’ ablation study in sec. 3.3, only using fast self-tuning background subtraction produces better results than CutMix. Why other methods even worse than CutMx? What’s the core reason for the improvement of using fast self-tuning background subtraction?
3.     The authors use batchsize=256, lr=0.1 for CIFAR training, while usually batchsize=128, lr=0.1 is used in previous works (Cutout). And as described in [1], the learning rate should be increased linearly with batchsize. This change of hyperparameters may need further explanation.

[1] Accurate, Large Minibatch SGD: Training ImageNet in 1 Hour.


============Post Rebuttal====================
After reading the feedback from authors, I still have my concerns. The novelty of this paper is too limited for ICLR. I really do not think a combination of CutMix with existing saliency detection method is a novel method. Moreover, the improvement over CutMix is diminishing. These main concerns are not addressed by the authors. So, my final recommendation is still rejection.

---

> ### Author Response · Authors · 2020-11-19
> **Response to Reviewer#2**
>
> - We thank the reviewer for the valuable feedback. We agree with the reviewer that the promotion is limited compared to CutMix. However, in image classification research domain, performance improvement by data augmentation has been gradually achieved. This is because the ImageNet dataset contains a huge number of images with much variety in both contents and texture characteristics as well as it has a total 1000 classes, which makes the improvement of performance difficult and highly saturated. However, the proposed method improves the model performance for image classification problem. Specifically, when the proposed method was applied without any traditional data augmentation, it significantly improved the classification performance of each model on every dataset as shown in Table 1 of the submitted manuscript. For instance, compared to the nearly competitive method CutMix, the proposed method improves the model performance by 1.85% and 5.69% for ResNet18, 2.35% and 6.76% for ResNet50 and 1.14% and 3.76% for WideResNet on CIFAR10 and CIFAR100 datasets, respectively.
>
>   Also, besides the performance improvement, the proposed method has proven itself to be effective on multiple datasets (CIFAR10, CIFAR100, and ImageNet) and tasks (image classification, object detection, and robustness against adversarially perturbed ImageNet classification).
>
> - Thanks for the valuable question. We think that saliency-based mixing strategy has pros and cons. That is, although saliency-based mixing can avoid picking up meaningless patches from source, it limits the available regions in images for data augmentation. That is, due to the center bias property of saliency regions, saliency-based mixing strategy may limit using the boundary regions which are away from the center salient regions for data augmentation. Therefore, if a saliency method itself does not have high confidence in estimating saliency regions, it may cause negative effect when being applied in data augmentation. However, we also guess that there are more intrinsic behaviors on saliency methods which boost the data augmentation performance.
>
> - Regarding the batch size and learning rate: We performed experiments on different batch size and learning rate for CIFAR training e.g., batch size 128 with learning rate 0.1, batch size 256 with learning rate 0.1 and batch size 256 with learning rate 0.25. The experimental results are presented in the Table-R4. It can be seen that the performance was similar for batch size 128 and 256 when the learning rate was 0.1. Also the performance was decreased when the learning rate was increased to 0.25. Based on the abovementioned experiments we decided to use learning rate of 0.1 and batch size of 256 to fully utilize the available resources. These results will be added to the supplementary material section.
>
> _Table-R4: Investigating the effect of batch size on CIFAR-10 dataset_
>
> | Method | Batch Size | Learning Rate | Run | Top-1 Error (%) |
> | :---: | :---: | :---: | :---: | :---: |
> ||||| CIFAR10+ |
> || 128 | 0.1 | 1 | 3.66 |
> |||| 2 | 3.48 |
> |||| 3 | 3.75 |
> |||| Average | **3.63** |
> |SaliencyMix| 256 | 0.1 | 1 | 3.55 |
> |||| 2 | 3.62 |
> |||| 3 | 3.75 |
> |||| Average | 3.65 |
> || 256 | 0.25 | 1 | 4.25 |
> |||| 2 | 4.29 |
> |||| 3 | 4.10 |
> |||| Average | 4.21 |

---

### Official Review · AnonReviewer1 · 2020-10-29
**Solid paper**

**Rating:** 9
**Confidence:** 5

**Review:**

The paper presents a method called SaliencyMix. They improve a method that augments images by adding random patches from other images. The innovation is that they select these patches using a saliency map.

This paper has an excellent discussion and critique of previous work. They discuss the existing work with a nice summary and then discuss reasons why selecting random patches can have issues. There is a clear argument for their method over selecting patches randomly. They make clear claims that this approach improves performance over randomly selecting patches. The experiments support this with sufficient related work (Cutout, Cutmix) and exploration of other design decisions and aspects about the idea.

The idea is relatively straightforward and is inline with existing literature. The paper is well executed so there is not much to complain about.

The availability of the source code is not clear from the text.

A potentially interesting analysis (but not required) is an analysis of the increased runtime in practice.

---

> ### Author Response · Authors · 2020-11-19
> **Response to Reviewer#1**
>
> - We thank the reviewer for the appreciation. Following the reviewer’s suggestion, we updated our manuscript by providing the Github link of our source code. For the convenience of the reviewer we show the link here also  (https://github.com/SaliencyMix/SaliencyMix).
>
> - We want to clarify that the proposed data augmentation technique is only applied during the training process and as a result it does not affect the inference time.

---

### Official Review · AnonReviewer4 · 2020-11-07
**Simple and useful idea, well presented.**

**Rating:** 7
**Confidence:** 4

**Review:**

*Summary and contributions:*
This paper proposes a new data augmentation strategy to train image classifiers and object detectors. The key insight is to use an image saliency signal to guide where to crop-and-paste images when mixing them. The paper includes an exploration of the design space of such approach, and multiple experimental results showing the empirical superiority of the proposed approach compared to existing data augmentation strategies.


*Significance:*
The paper is interesting because it provides a new trick in the bag of tricks that is both simple to understand, reasonable to argue for, and (now) has good empirical support (for classification, detection, and adversarial attack robustness).


Originality:
Limited. Although no previous work provides the experimental results presented here, the results are expected. This work is good A+B incremental work.


*Strengths:*
* Overall the paper read easily. The general argumentation, method description, and experiments are all reasonably well described.
* Simple ideas that provide good results. In retrospectively it might seem obvious, yet not explored before.
* Widely applicable for all methods using pre-trained image classifiers.

*Weaknesses:*
* Although the experimental section is good, some elements are missing. E.g. adding non-data-augmented as reference point in the plot, or considering CAM as a saliency strategy.
* Some sections of the text would benefit from revisiting the English.
* The method implicitly relies on having “simple images” with one dominant foreground object  like the ones in CIFAR and ImageNet. The saliency / “object of interest”  detection method depends on these characteristics. Ideally the paper would be more upfront on these assumptions. Especially in the context of the “Rethinking ImageNet Pre-training”, ICCV 2019 work.
* The related work section for saliency is very partial.
* Some of the saliency methods evaluated use training data, even the ones that do not have been tuned using additional data. The paper would benefit from a discussion of this additional information.


*Correctness:* Paper seems correct. There are minor shortcomings in the experimental protocol, but nothing that would foreseen invalidating the main conclusions of the paper.

*Clarity:* Paper presentation is clear.






*Relation to prior work:*
Related work section has a reasonable extent.
Regarding data augmentation, the paper compares with the main methods.
The text mentions Lemley 2017, however I think it would be also worth mentioning AutoAugment Cubuk 2019; and justifying why it is not included in the results comparison.

The saliency detection is very partial, and does not cover the main works in the area.
For one the text does not clarify “which kind” of saliency is considered (where will a human look ? which are the main objects of the scene ? which is the main object of the scene ?). Depending on which one, discussing the main performers in the related benchmarks (e.g. table in Qin 2019 paper) seem relevant.
From what I grasp, the proposed method would actually want to have as input a weakly supervised class-conditional segmentation. And “saliency” is used as a proxy for this. Discussing the relation to (image-level labels) weakly supervised segmentation would also be welcome.
(I would guess it could provide even better results, but at a much higher computational and system complexity cost). In particular CAM is discussed in section 4.3. Would that not be a task-specific way to obtain the desired “semantically important regions”  ?


When preparing the camera ready, please consider discussing the concurrent work of https://arxiv.org/pdf/2009.06962.pdf which seems related (seems recent enough, ICML 2020, to give the benefit of presumed concurrency).


*Reproducibility:*
The overall algorithm is simple to understand and re-implement.
The selected saliency method “Wand and Dudek, 2014” seems to be a video saliency method. From a quick inspection of that paper it is not immediately clear to me how to transpose it for single image saliency. Since this method is used in most experiments, providing more details of the implementation and its parameters are necessary to reproduce the key results.


*Specific per-section feedback:*

Section 1:
- every field: is too broad of a statement, remove/rephrase the first sentence.
- extremely complex -> complex
- generalizability -> generalization
- undesired to the CNN since -> undesired since
- does not allow … to have any uniformative pixel: double negative, consider simplifying.
- semantically important region: these are task dependent. How do you ensure the saliency to match the task / semantics ?


Section 2.1:
- comes into account that aims: unclear, please rephrase.


Section 2.2:
- unable to … 1 or 0: this is not true, please rephrase.
- intermediate values: you mean closer to 0.5 ?
- Thereby, helps: unsure if this is proper English.


Section 2.3: See comments above regarding related work.
- Which kind of saliency is considered here ? Which are the relevant benchmarks ? Which training data / evaluation data are these methods bringing (indirectly) into the system ?
- BAsNet for example, was trained on 10k images. Why not simply include these (and their mask) as part of the pretraining when considering some of the baselines ?
Section 3.1:
- selected training image -> selected training (source) image. In general do give hints for the subscript meaning of I_s, I_vs, etc.
- Why only one pixel with maximum intensity ? What happens if (due to quantization) two pixels have the same value ?
This is clarified later in the text, but some context would be welcome in the mention here.
Figure 3:
- Add non-augmented result bar as reference point.
- Consider showing the five points per bar, or some hint for the variance in these results. From the plot it is left unclear if the fluctuations across methods are significant or minor (since not reference point, nor sense of the variance).
Section 3.3:
- Tiny-Imagenet: saliency methods tend to _not_ be scale invariant (in particular trained methods like BasNet). How is this handled ? Why would one expect Tiny-Imagenet to provide conclusive data ?
- What about CAM, or any other class-conditioned saliency / weakly supervised segmentation method ?
Section 3.4:
- found out -> consider
- What about “Salient to Random“ ? That seems a reasonable option too ?
- are identical -> are similar


Section 4.1:
- SOTA top-1 error: there are 20 methods that claim better results in https://paperswithcode.com/sota/image-classification-on-cifar-10 and https://paperswithcode.com/sota/image-classification-on-cifar-100 that claim better results. Maybe temperate the “SOTA” claim, with phrasings like “best known results for model X”  or similar. One specific model result, far from best known, does not constitute “state of the art”  in my understanding.


Section 4.2:
- Because in -> This is because in


*Updates after reviews and authors feedback:*

The updates from the author are appreciated and make the arguments of the paper clearer.
After reading the other reviews and discussions, I have downgraded my score to "7: Good paper, accept".

(Please note that in the current pdf the table at the top of page 7 has formatting issues.)

---

> ### Author Response · Authors · 2020-11-19
> **Response to Reviewer#4**
>
> **We thank the reviewer for the effort and the appreciation.**
>
> - Following the reviewer’s suggestion, we added the non-data-augmented results as a reference point to the plot in Figure 3 (a) and 3 (b) of the revised manuscript.
> - Also, based on the reviewer’s suggestion, we investigated the effect of CAM as a saliency strategy when incorporated in the proposed data augmentation technique. We performed experiments on CIFAR10 with ResNet18. The pre-trained baseline model is used to extract the CAM. The results are reported from the average of five runs and presented in Table-R3. Due to the space limitation in the manuscript, we will add this result in the supplementary material.
>
> _Table-R3: Verifying the effect of CAM as a saliency model for the proposed data augmentation method._
>
> | Method | Run | Top-1 Error (%) |
> | :---: | :---: | :---: |
> ||| CIFAR10+ |
> | | 1 | 4.44 |
> |SaliencyMix| 2 | 4.76 |
> |(CAM as a Saliency Model)| 3 | 4.66 |
> || 4 | 4.48 |
> || 5 | 4.63 |
> || Average | 4.59 |
>
> - According to the reviewer’s suggestion, we’ll revise the English by some native English speaker during the camera ready submission.
> - We agree that most saliency methods implicitly rely on having “simple images” with one dominant foreground object like the ones in CIFAR and ImageNet. The saliency / “object of interest” detection method depends on these characteristics. As our future direction, we plan to extend our work by integrating Mosaic, a decent data augmentation method for image detection where Saliency information is locally calculated and applied to elaborate Mosaic strategy.
> - Based on the reviewer’s recommendation, we updated the related work section by providing a brief history about saliency detection in the revised manuscript.
> - We agree with the reviewer that additional or high-quality data may help increase in saliency estimation for learning-based saliency methods. In our work, we used Montabone’s saliency method [Ref3] which is an algorithmic based one and does not require training. However, we believe that training based saliency methods can also be applied where the quality and quantity of data for training the saliency methods may be correlated with the effectiveness of data augmentation.
>
> - Relation to prior work:
>   - Based on the reviewer’s suggestion, AutoAugment [Ref1] and  PuzzleMix [Ref2] are added to the related work section and in the result comparison.
>   -Regarding the perspective that ‘Saliency information’ is proxy semantic information (e.g., partial information of semantic segmentation), we agree the reviewer’s view point, and think that such interpretation for SaliencyMix can give a good direction for future study. According to the reviewer’s comment, we added this perspective and interpretation in Conclusion of the revised manuscript. Also, thanks to the reviewer’s comment, we will consider utilizing more detailed and/or high level semantic information for data augmentation as future work.
>   - We agree with the reviewer that the CAM visualization is task specific. Actually, we wanted to verify and compare the localization ability of models trained with different data augmentation techniques. Based on the reviewer’s concern, beside the CAM visualization on augmented images, we newly added CAM visualization results for different data augmentation techniques on un-augmented images in Figure 4 of the revised manuscript.
>
>
> - Reproducibility:
>   We first apologize for mentioning an incorrect reference of the used saliency model in the paper. It was mistake to mention “Fast Self Tuning Background Subtraction” as our main saliency model. Actually, we used OpenCV fine-grained static saliency detection model proposed by S. Montabone and A. Soto (denoted as Montabone’s method) [R3] for our SaliencyMix. To make it clear and reproducible, we release our source code in Github (link: https://github.com/SaliencyMix/SaliencyMix)
>
> - Specific per-section feedback:
> We again sincerely express our thanks to the reviewer for the valuable comments on the revision of our paper. We exerted all possible efforts to answer the questions and comments of the reviewers, and wish that the reviewers and readers could better understand the originality and novelty of our works from this revised manuscript.
>
> References
>
> [Ref1]. E. D. Cubuk, B. Zoph, D. Mané, V. Vasudevan and Q. V. Le, "AutoAugment: Learning Augmentation Strategies From Data,"  _2019 IEEE/CVF Conference on Computer Vision and Pattern Recognition (CVPR)_, Long Beach, CA, USA, 2019, pp. 113-123, doi: 10.1109/CVPR.2019.00020.
>
> [Ref2]. Jang-Hyun Kim, Wonho Choo and Hyun Oh Song, “Puzzle Mix: Exploiting Saliency and Local Statistics for Optimal Mixup”, _International Conference on Machine Learning (ICML)_, 2020.
>
> [Ref3]. Sebastian Montabone and Alvaro Soto. Human detection using a mobile platform and novel features derived from a visual saliency mechanism. _In Image and Vision Computing_, Vol. 28, no. 3, pages 391–402. Elsevier, 2010.

---

### Official Review · AnonReviewer5 · 2020-11-07
**Interesting approach; some baselines would be useful**

**Rating:** 7
**Confidence:** 4

**Review:**

This paper proposes an improvement on the cutmix strategy of data augmentation, where the source patch is selected not randomly but based on saliency. Results show improvements w.r.t mixup and other related strategies on Imagenet, CIFAR 10/100 and also transfer to object detection

Pros:
- The approach is intuitive and makes sense (which is more than can be said for the baselines of CutMix and MixUp). I think this approach probably starts to get to the heart of why these previous strategies work: they are probably less effective ways of doing what this paper suggests.
- The results seem quite promising, and the improvements seem significant.

Cons:
- I am a bit surprised that the best strategy is Sal + Corr. I understand the author's reasoning, but I find it strange that it has that big of an effect. If the author's reasoning is correct, I would assume that a random placement of the cut patch would be just as effective. Could the authors try Sal + Random instead?
- It is well known in the saliency literature that saliency has a center bias. This suggests a baseline where the source patch is always cut from the center. I would suggest the authors add this baseline.
- I am not sure about the point of using CAM visualizations on augmented images. Perhaps a better visualization might be CAM visualization of the models trained with each kind of visualization on the unaugmented images?

---

> ### Author Response · Authors · 2020-11-19
> **Response to Reviewer#5**
>
>  **We thank the reviewer for the appreciation and valuable comments.**
> - Following the reviewer’s suggestion, we have performed experiments to check the effect of “Salient to Random” mixing strategy. We have used ResNet18 as a baseline and trained our proposed method with “Salient to Random” mixing strategy on CIFAR10. The results are reported from the average of five runs. Table-R1 presents the experimental results including the previously found results for SaliencyMix with “Salient-to-Corresponding” strategy. It can be seen that the performance for both the cases are almost similar.
>
> _Table-R1: Performance comparison of the proposed method when applied with "Salient-to-Random" and "Salient-to-Corresponding" mixing strategy. Experiments are performed on CIFAR10 using ResNet18 architecture._
>
> |Method| Run | Top-1 Error (%) |
> | :---: | :---: | :---: |
> |     |     | CIFAR10+ |
> | | 1 | 3.81 |
> |SaliencyMix| 2 | 3.63 |
> |(Salient-to-Random)| 3 | 3.66 |
> || 4 | 3.57 |
> || 5 | 3.53 |
> || Average | 3.64 |
> | | 1 | 3.55 |
> | SaliencyMix| 2 | 3.62 |
> |(Salient-to-Corresponding)| 3 | 3.75 |
> || 4 | 3.64 |
> || 5 | 3.69 |
> || Average | 3.65 |
>
> - We agree with the reviewer that the saliency has a center bias. Following the reviewer’s suggestion, we performed experiments where the source patch is always cut from and mix to the center. Table-R2 presents the experimental results. Center-to-Center approach achieves top1-1 error of 4.45% and 38.27% on CIFAR10 and Tiny-ImageNet, respectively. While the proposed strategy achieves top-1 error of 3.65% and 36.03% on CIFAR10 and Tiny-ImageNet, respectively. The results suggest that if the source patch is always cut from the center and mixed to the center, it restricts the model from generating diverse samples. According to the reviewer’s comment, we added this baseline (Center-to-Center) in Figure 3 of the revised manuscript.
>
> _Table-R2: Performance checking of the proposed method when the source patch is always cut from and mixed to the center._
>
> | Method | Run | Top-1 Error (%) | Top-1 Error (%) |
> | :---: | :---: | :---: | :---: |
> ||| CIFAR10 | Tiny-ImageNet |
> || 1 | 4.49 | 37.43 |
> |SaliencyMix | 2 | 4.46 | 38.03 |
> |(Center-to-Center)| 3 | 4.47 | 39.36 |
> || 4 | 4.40 | 38.67 |
> || 5 | 4.43 | 38.86 |
> || Average | 4.45 | 38.27 |
>
> - According to the reviewer’s suggestion, we added the CAM visualization of the models trained with different augmentation techniques on un-augmented images in Figure 4 of the revised manuscript, where all the images are randomly taken from ImageNet validation set. The visualization results show that the proposed SaliencyMix tends to help increase in localization performance as the CAM maps focus more on target objects (e.g., Mountain Tent, Marmot, and Vizsla etc.).

---

### Public Comment · ~June_Yong_Yang1 · 2020-11-17
**Comparison with existing saliency-based mixup methods(PuzzleMix, AttentiveCutmix)?**

I wonder what is the key methodical difference between this work and AttentiveCutmix [1]?
Also I think a comparison with existing best saliency-based mixup algorithms such as PuzzleMix [2] and AttentiveCutmix [1] would be needed.

[1] "Attentive Cutmix: An Enhanced Data Augmentation Approach for Deep Learning Based Image Classification", Walawalkar et al. ICASSP 2020 (https://arxiv.org/abs/2003.13048)
[2] "Puzzle Mix: Exploiting Saliency and Local Statistics for Optimal Mixup". Kim et al. ICML 2020 (https://arxiv.org/abs/2009.06962)

---

> ### Author Response · Authors · 2020-11-20
> **About the comparison with PuzzleMix and Attentive-CutMix**
>
> We thank Mr. Yang for showing interest in this work.
>
> - Regarding the similarity with Attentive-CutMix
>
>   Both the Attentive-CutMix and our paper is motivated from the same perspective but with different approaches.
>
>   Attentive-CutMix first extracts the feature map of the source image by passing it to a pre-trained network and selects few important points of the feature map. By relating back those locations (down-sampled feature map) to the original image, the patches are then selected. It is to be noted that the selected patches are spatially disconnected. However, spatial relation is important to classify an image or object.
>
>   Unlike Attentive-CutMix, we extract saliency map of source image and then select the most important region to mix with target image. Since a single patch is selected, the spatial relation is maintained that helps the convolutional neural network (CNN) to learn a better feature representation.
>
>   We also would like to inform you that our paper was submitted to ICML 2020 which supports the originality of our work.
>
> - Regarding the result comparison
>
>   Please be informed that we already presented the result comparison with PuzzleMix in the rebuttal version of our paper.
>
>   But we noticed few issues about the Attentive-CutMix results as follows:
>   - Experimental setup of Attentive-CutMix is different from the standard setup. For example, the network was trained for 80 and 120 epochs for CIFAR10 and CIFAR100 datasets, respectively, while other methods usually trained for 200 epochs. We afraid of that the results are reported before the convergence of the networks.
>   - Batch size of 32 was used with a learning of 0.001, while other standard methods used batch size of 128 with a learning rate of 0.1.
>   - Also, the source code is not publicly available. As a result, we could not perform experiments using Attentive-CutMix with the standard setup for a fair comparison with the proposed method. Further, ImageNet classification results are not reported in Attentive-CutMix paper.

---

### Comment · ~JangHyun_Kim1 · 2021-01-15
**Comments for ImageNet experiments (Puzzle Mix)**

Hello! I'm the one of the authors of Puzzle Mix.

First, I'd like to congratulate the acceptance. Because the method is highly related to ours it was interesting to read your paper. However, I want to give some comments for Puzzle Mix ImageNet results (which is added in the rebuttal process).

The reported performance of Puzzle Mix (22.49%) is from a model trained with 100 epochs. When we train a model using Puzzle Mix for **300 epoch** as the same as CutMix paper, we get **21.24%** Top-1 error rate (5.71% for Top-5) which is comparable to your results.

Please refer to our updated version (https://arxiv.org/abs/2009.06962) and codes (https://github.com/snu-mllab/PuzzleMix/tree/master/cutmix). You can download and test the Puzzle Mix ImageNet 300 epoch model. It will be appreciate if you update the table following our results for a fair comparison.

Best,
Jang-Hyun

---

### Author Response · Authors · 2021-01-19
**Acknowledgement on Acceptance**

The authors thank all the personnel related with ICLR-2021. Special thanks to the reviewers who really worked hard and gave thoughtful review comments to improve the manuscript quality. Also thanks to the open-review team who made this easy. Finally, special thanks to the chairs for organizing such a mega event.

---

### Decision · Program_Chairs · 2021-01-07
**Final Decision**

**Decision:**

Accept (Poster)

**Comment:**

This paper proposes an approach to data augmentation to train image recognition models called SaliencyMix, which involves pasting salient regions (as judged by some saliency detector) of one image into another, and mixing the two labels accordingly. Most reviewers generally agreed that the proposed approach is simple -- it is easy to understand the method and its motivation, especially in the context of related augmentation approaches like CutMix -- and has solid experimental results demonstrating its effectiveness.

The main objection the more negative reviewers had to the work is a perceived lack of novelty. In my view it is a new method (even if similar to prior work like CutMix), and as AR5 argues: "this approach probably starts to get to the heart of why these previous strategies work: they are probably less effective ways of doing what this paper suggests." The improvements in Table 1, columns 1 & 3 (CIFAR-10 & CIFAR-100) especially speak to this -- these improvements with traditional augmentation disabled are quite substantial, even though the differences become marginal when moving to the "+" augmented versions of the dataset (as well as in ImageNet). So although the method is indeed similar to CutMix, I agree that it offers valuable insight into why these previous methods work. Besides which the results *do* show improvements over similar methods, even if the improvements are marginal.

Overall, I recommend accepting the paper as it provides useful insight into why prior methods work and proposes a new one that in practice works slightly better. Minor comments for the camera-ready version:

Please revise the writing based on AR4's good suggestions.

Highlighting a comment from AR4:
> BAsNet for example, was trained on 10k images. Why not simply include these (and their mask) as part of the pretraining when considering some of the baselines ?

I recommend including discussion of this important point in the final version of the paper. The learning-based approaches are effectively using additional training data. It's good that a non-learning-based method happens to perform best so that the results remain comparable with prior work, but this should nonetheless be discussed if the learning-based approaches are to be included.

Please remove the blue text coloring (if not already planned -- I'm not sure if this was done as a "diff" for the response).

> Figure 3(a-b) show that Montabone & Soto (2010) performs better on both the datasets and the effects are identical on CIFAR10 and ImageNet datasets

I do not see how Figure 3 shows this. Is "OpenCV Saliency" in the figure using the method from Montabone & Soto (2010)? Please clarify this by making the connection between the bar labels in the figure and the discussion in the text clear for the camera-ready version of the paper.

---

> ### Author Response · Authors · 2021-01-19
> **Thanks and Response to the Final Decision**
>
> **First of all, we thank the PC for appreciating and accepting our work to present in ICLR-2021. Also the authors are grateful for the PC's valuable suggestions.**
>
> *Concern-1:* Please revise the writing based on AR4's good suggestions.
> - Response:  The authors thank the PC for valuable suggestions. It should be noted that we have reflected most of the reviewer’s comments in the rebuttal version of our paper. And we'll reflect the remaining comments in the final camera-ready version. Also we'll revise the manuscript carefully during camera-ready submission.
>
> *Concern-2:* Please remove the blue text coloring (if not already planned -- I'm not sure if this was done as a "diff" for the response).
> - Response: The authors thank the PC for pointing out every details. Actually, the blue text was used to highlight the changes during rebuttal period. We'll remove the blue text in the final version of our manuscript.
>
> *Concern-3:* Clarification about "OpenCV Saliency" and "Montabone & Soto" as saliency detection method.
> - Response: Yes, OpenCV Saliency detection method uses "Montabone and Soto" method. We are sorry for creating such confusion. We'll clarify it and also modify Figure 3 by replacing "OpenCV Saliency" with "Montabone & Soto" in the final version of our paper.